# Effects of Stoichiometry on Structural, Morphological and Nanomechanical Properties of Bi$_2$Se$_3$ Thin Films Deposited on InP(111) Substrates by Pulsed Laser Deposition

**Yeong-Maw Hwang [1], Cheng-Tang Pan [1], Bo-Syun Chen [1], Phuoc Huu Le [2,*], Ngo Ngoc Uyen [2], Le Thi Cam Tuyen [3], Vanthan Nguyen [4], Chih-Wei Luo [5], Jenh-Yih Juang [5], Jihperng Leu [6] and Sheng-Rui Jian [7,8,*]**

[1] Department of Mechanical and Electro-Mechanical Engineering, National Sun Yat-Sen University, Kaohsiung 804, Taiwan; ymhwang@mail.nsysu.edu.tw (Y.-M.H.); pan@mem.nsysu.edu.tw (C.-T.P.); bosyun815@gmail.com (B.-S.C.)

[2] Department of Physics and Biophysics, Faculty of Basic Sciences, Can Tho University of Medicine and Pharmacy, 179 Nguyen Van Cu Street, Can Tho City 94000, Vietnam; nnuyen@ctump.edu.vn

[3] Faculty of Basic Sciences, Nam Can Tho University, 168 Nguyen Van Cu (Ext) Street, Can Tho City 94000, Vietnam; ltctuyen89@gmail.com

[4] Department of Electrical and Electronic Engineering, Faculty of Automotive Engineering, Ngo Quyen University, Thu Dau Mot City 820000, Vietnam; nguyenvanthan1010@gmail.com

[5] Department of Electrophysics, National Chiao Tung University, Hsinchu 300, Taiwan; cwluo@mail.nctu.edu.tw (C.-W.L.); jyjuang@g2.nctu.edu.tw (J.-Y.J.)

[6] Department of Materials Science and Engineering, National Chiao Tung University, Hsinchu 300, Taiwan; jimleu@mail.nctu.edu.tw

[7] Department of Materials Science and Engineering, I-Shou University, Kaohsiung 840, Taiwan

[8] Department of Fragrance and Cosmetic Science, Kaohsiung Medical University, 100 Shin-Chuan 1st Road, Kaohsiung 80782, Taiwan

\* Correspondence: lhuuphuoc@ctump.edu.vn (P.H.L.); srjian@gmail.com (S.-R.J.); Tel.: +886-7-657-7711 (ext. 3130) (S.-R.J.)

**Abstract:** In the present study, the structural, morphological, compositional, nanomechanical, and surface wetting properties of Bi$_2$Se$_3$ thin films prepared using a stoichiometric Bi$_2$Se$_3$ target and a Se-rich Bi$_2$Se$_5$ target are investigated. The Bi$_2$Se$_3$ films were grown on InP(111) substrates by using pulsed laser deposition. X-ray diffraction results revealed that all the as-grown thin films exhibited were highly *c*-axis-oriented Bi$_2$Se$_3$ phase with slight shift in diffraction angles, presumably due to slight stoichiometry changes. The energy dispersive X-ray spectroscopy analyses indicated that the Se-rich target gives rise to a nearly stoichiometric Bi$_2$Se$_3$ films, while the stoichiometric target only resulted in Se-deficient and Bi-rich films. Atomic force microscopy images showed that the films' surfaces mainly consist of triangular pyramids with step-and-terrace structures with average roughness, $R_a$, being ~2.41 nm and ~1.65 nm for films grown with Bi$_2$Se$_3$ and Bi$_2$Se$_5$ targets, respectively. The hardness (Young's modulus) of the Bi$_2$Se$_3$ thin films grown from the Bi$_2$Se$_3$ and Bi$_2$Se$_5$ targets were 5.4 GPa (110.2 GPa) and 10.3 GPa (186.5 GPa), respectively. The contact angle measurements of water droplets gave the results that the contact angle (surface energy) of the Bi$_2$Se$_3$ films obtained from the Bi$_2$Se$_3$ and Bi$_2$Se$_5$ targets were 80° (21.4 mJ/m$^2$) and 110° (11.9 mJ/m$^2$), respectively.

**Keywords:** Bi$_2$Se$_3$ thin films; nanoindentation; hardness; pop-in; surface energy

## 1. Introduction

Bismuth selenide ($Bi_2Se_3$) is of great interest owing to its intriguing physical properties as a three-dimensional topological insulator [1–5], and potential applications in spintronics [6], optoelectronics [7] and quantum computation [8]. In addition, $Bi_2Se_3$ possesses excellent thermoelectric properties at room-temperature [9,10] and low temperature regimes [11]. For fundamental studies and application purposes, it is essential to grow $Bi_2Se_3$ thin films with high-quality and to have comprehensive characterizations of their physical properties, including the mechanical properties [12–14].

Nanoindentation is a versatile technique ubiquitously used to obtain the basic mechanical parameters, such as the hardness and elastic modulus, as well as to delineate the deformation mechanisms, creep and fracture behaviors of various nanostructured materials [15–18] and thin films [19–23] with very high sensitivity and excellent resolution. On the other hand, wettability is an important property of a solid surface, which is intimately related to the chemical compositions and morphology of the surface [24]. The peculiar wetting behaviors exhibited on the surface of two-dimensional and van der Waals layered materials have been receiving dramatically increased interest in recent years [25–27]. It implies that specific water–substrate interaction features are relevant to the atomic and electronic structures of the layered materials. In particular, the hydrophobic surface (water contact angle, $\theta_{CA} > 90°$) can be used in many applications of self-cleaning surfaces and antifogging [28,29]. Consequently, how to control the behavior of hydrophobicity or hydrophilicity of films' surfaces is also of great importance in realizing the designed functionality for device applications.

Because of the high volatility of selenium (Se), $Bi_2Se_3$ tends to form Se vacancies or antisites that serve as donors to result in a sufficiently high carrier concentration and low carrier mobility [30,31]. When severe loss of Se-atoms occurs during the thin-film growth at elevated substrate temperatures, pure phase $Bi_2Se_3$ film is usually not achieved, and the obtained films may present impurity phases or even turn into another phase [32]. Thus, to overcome this problem and obtain high-quality stoichiometric $Bi_2Se_3$ thin-films, a Se-rich environment is necessary during films' growth. Indeed, this strategy has been employed to grow high-quality $Bi_2Se_3$ thin films by creating a Se-rich environment with a Se:Bi flux ratio ranging from 10:1 to 20:1 using molecular beam epitaxy (MBE) [33,34]. Pulsed laser deposition (PLD) offers a high instantaneous deposition rate, relatively high reproducibility, and low costs. The PLD has been used for growing epitaxial and polycrystalline $Bi_2Se_3$ thin films [9,30,35–37]. In 2011, Onose et al. [35] successfully grew epitaxial $Bi_2Se_3$ thin-films on InP(111) substrates using a designed target with an atomic ratio of Bi:Se of 2:8. Yet, systematic investigations on the effects of target composition, and hence the resultant films' stoichiometry, on the properties of $Bi_2Se_3$ thin films have been relatively scarce.

Herein, we conducted comprehensive characterizations of the structural, compositional, morphological, nanomechanical, and wetting properties of $Bi_2Se_3$ thin films grown on InP(111) substrates by PLD. In particular, two different targets (i.e., a stoichiometric target of $Bi_2Se_3$ and a Se-rich target of $Bi_2Se_5$) were deliberately used to tune the stoichiometry of the resultant $Bi_2Se_3$ films and to unveil its effects on the surface wettability and nanomechanical properties, since both characteristics are of pivotal importance for their practical applications in $Bi_2Se_3$ thin film-based microelectronic and spintronic devices.

## 2. Materials and Methods

In order to study the effects of film stoichiometry, two targets with different composition effects were used. One is stoichiometric $Bi_2Se_3$ and another is a Se-rich target with a nominal composition of $Bi_2Se_5$. The targets were purchased from Ultimate Materials Technology Co., Ltd. (Ping-Tung City, Taiwan). Noticeably, though having differences in Se/Bi atomic ratios of 3/2 and 5/2, both $Bi_2Se_3$ and $Bi_2Se_5$ targets were polycrystalline with the right $Bi_2Se_3$ phase. $Bi_2Se_3$ thin films were deposited on InP(111) substrates using PLD at a substrate temperature of 350 °C in vacuum at a base pressure of $4 \times 10^{-6}$ Torr (~0.53 mPa). For the PLD process, ultraviolet (UV) pulses (20-ns duration) from a KrF

excimer laser ($\lambda$ = 248 nm, repetition: 1 Hz) were focused on the polycrystalline $Bi_2Se_3$ or $Bi_2Se_5$ target at a fluence of 5.5 J/cm$^2$. The target-to-substrate distance was 40 mm. The target was ablated for approximately 5 min in order to clean its surface before every deposition. The deposition time was 25 min, which resulted in an average $Bi_2Se_3$ film thickness of approximately 191 nm (the growth rate of approximately 1.27 Å/pulse).

The crystal structure and surface morphology of the $Bi_2Se_3$ thin films were characterized by X-ray diffraction (XRD; Bruker D8, CuK$\alpha$ radiation, $\lambda$ = 1.5406 Å, Bruker, Billerica, MA, USA) and field emission scanning electron microscopy (SEM, JEOL JSM-6500, JEOL, Pleasanton, CA, USA) operated at an accelerating voltage of 15 kV, respectively. Film compositions were analyzed through Oxford energy-dispersive X-ray spectroscopy (EDS, Inca X-sight 7558, Oxford Instruments plc., Oxfordshire, UK) equipped with the SEM instrument at an accelerating voltage of 15 kV, dead time of 22–30%, and collection time of 60 s. The atomic percentage of each film was determined by averaging the values measured in 5 or more distinct $14 \times 20$ $\mu m^2$ areas on the surface of films. Moreover, the surface morphology and roughness of the thin films were examined using atomic force microscopy (AFM; Veeco Escope, Veeco, New York, USA).

The nanoindentation was performed on a Nanoindenter MTS NanoXP$^{\circledR}$ system (MTS Cooperation, Nano Instruments Innovation Center, Oak Ridge, TN, USA) with a pyramid-shaped Berkovich diamond tip. The nanomechanical properties of the $Bi_2Se_3$ thin films were measured by nanoindentation with a continuous contact stiffness mode (CSM) [38]. At least 20 indentations were performed on each sample and the distance between the adjacent indents was kept at least 10 $\mu m$ apart to avoid mutual interferences. We also followed the analytic method proposed by Oliver and Pharr [39] to determine the hardness and Young's modulus of measured materials from the load–displacement results. Thus, the hardness ($H$) and Young's modulus ($E$) of the $Bi_2Se_3$ thin films are obtained and the results are listed in Table 1. Moreover, the surface wettability of the $Bi_2Se_3$ thin films under ambient conditions was monitored using a Ramehart Model 200 contact angle goniometer (Ramé-hart, Succasunna, NJ, USA) with deionized water as the liquid.

**Table 1.** The microstructural parameters, nanomechanical properties, contact angle and surface energy of $Bi_2Se_3$ thin films. The mechanical properties of InP(111) are also listed.

| Sample | $D$ (nm) | $R_a$ (nm) | $H$ (GPa) | $E$ (GPa) | $\tau_{max}$ (GPa) | $\theta_{CA}$ | $(\gamma^d)_s$ (mJ/m$^2$) |
|---|---|---|---|---|---|---|---|
| $Bi_2Se_3$ thin film on InP(111) substrate ($Bi_2Se_3$ target) | 29.7 | 2.41 | 5.4 | 110.2 | 1.8 | 80° | 21.4 |
| $Bi_2Se_3$ thin film on InP(111) substrate ($Bi_2Se_5$ target) | 26.0 | 1.65 | 10.3 | 186.5 | 3.4 | 110° | 11.9 |
| $Bi_2Se_3$ thin film on sapphire substrate [14] | 34.2 | 8.5 | ~2.1 | ~58.6 | ~0.7 | — | — |
| Single-crystal $Bi_2Se_3$ [13] | — | — | ~0.4–0.9 | ~2–9 | — | — | — |
| Single-crystal InP(111) [40] | — | — | ~5 | 72.4–76.2 | 1.96 | — | — |

## 3. Results and Discussion

### 3.1. Structural and Morphological Properties

$Bi_2Se_3$ has a rhombohedral structure with a space group $D_{3d}^5 (R\bar{3}m)$ that can be described by a hexagonal primitive cell with three five-atomic-layer thick lamellae of –($Se^{(1)}$–Bi–$Se^{(2)}$–Bi–$Se^{(1)}$)–, in which the atomic layers are stacked in sequence along the *c*-axis [9]. The XRD patterns of the $Bi_2Se_3$ thin films obtained from the $Bi_2Se_3$ and $Bi_2Se_5$ targets are shown in Figure 1. As is evident from Figure 1, besides the diffraction peaks of InP substrates at 26.3° and 54.1° (JCPDS PDF#00-032-0452), the films exhibited highly *c*-axis-preferred orientation with (006), (0015), and (0021) diffraction peaks of the $Bi_2Se_3$ phase (JCPDS PDF#33-0214). However, minor diffraction peaks belonging to the BiSe phase (PDF#29-0246) can be identified. It is noticed that, although both of the as-grown films exhibit highly *c*-axis preferred orientation of the $Bi_2Se_3$ phase, a slight relative shift in diffraction angles indicative of modification of the *c*-axis parameter is observed. Indeed, by using the dominant $Bi_2Se_3$ (006) and $Bi_2Se_3$ (0015) peaks and the hexagonal unit cell relationship [32], the average *c*-axis lattice constant of

the $Bi_2Se_3$ thin films prepared using $Bi_2Se_3$ and $Bi_2Se_5$ targets were 28.39 Å and 28.25 Å, respectively, whose values were slightly smaller the *c*-axis lattice constant of 28.63 Å from the database of $Bi_2Se_3$ powder (JCPDS PDF#33-0214). This could be due to the difference in the internal stress built up during the deposition.

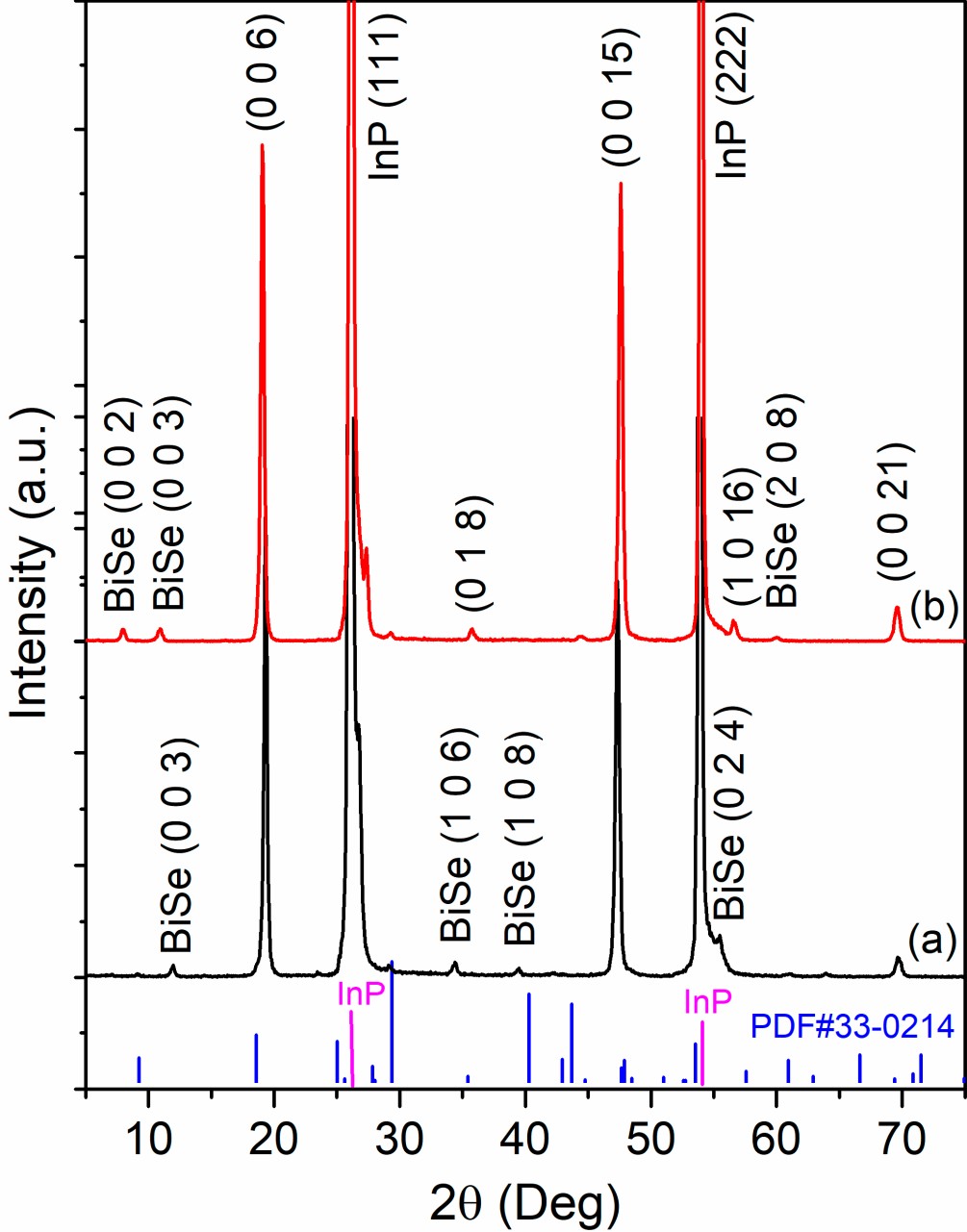

**Figure 1.** XRD patterns of $Bi_2Se_3$ thin films grown on InP (111) substrates from two different targets of $Bi_2Se_3$ (**a**) and $Bi_2Se_5$ (**b**) using pulsed laser deposition.

The grain sizes (*D*) of the $Bi_2Se_3$ films were estimated using the Scherrer equation $D = 0.9\lambda/\beta\cos\theta$, where $\lambda$, $\beta$, and $\theta$ are the X-ray wavelength, full width at half maximum of the $Bi_2Se_3$ (006)-oriented peak, and Bragg diffraction angle, respectively. The estimated *D* values of the $Bi_2Se_3$ thin films prepared using $Bi_2Se_3$ target and $Bi_2Se_5$ target were 29.7 nm and 26.0 nm, respectively.

Figure 2 shows the AFM and SEM-EDS results of $Bi_2Se_3$ thin films prepared using the $Bi_2Se_3$ and $Bi_2Se_5$ target, respectively. As shown in Figure 2a,b, the films mainly consist of triangular pyramids

with features of step-and-terrace structures. This is a clear indication that the films are growing along the [0001] direction, which is consistent with XRD results displayed in Figure 1. The films also exhibit highly smooth surfaces with the centerline average roughness $R_a$ being ~2.41 nm and ~1.65 nm for films grown from the $Bi_2Se_3$ target and from the $Bi_2Se_5$ target, respectively. In addition, the films grown from the $Bi_2Se_5$ target also show clearer step-and-terrace structures with fewer large particle-like outgrowth defects on the surface as compared to the film grown from the $Bi_2Se_3$ target (see 3D images), indicating that these films are closer to the stoichiometric composition and, thus, are less defective.

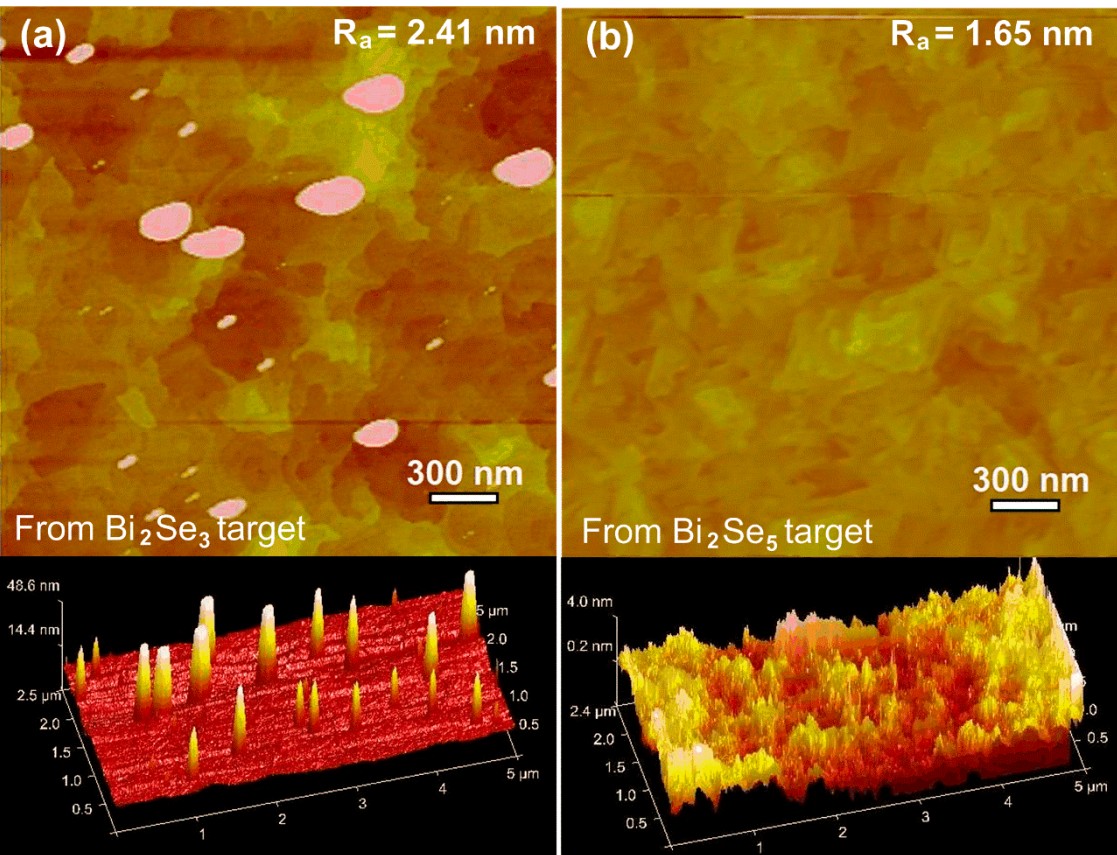

**Figure 2.** 2D and 3D AFM images of the $Bi_2Se_3$ thin films deposited from (**a**) $Bi_2Se_3$ target and (**b**) $Bi_2Se_5$ target.

The top-view SEM images displayed in Figure 3a,b further confirmed the aforementioned surface morphology. The cross-sectional view images shown at the bottom of Figure 3a,b indicate that the films are rather uniform with their thickness being in the range of 185~197 nm. Furthermore, as is evident from the EDS results displayed in the insets of Figure 3a,b and the typical EDS spectra of the corresponding thin films shown in Figure 3c, the composition of the film prepared from the $Bi_2Se_3$ target clearly showed a substantial Se-deficiency of about 4.4 at.%, while the film prepared from the $Bi_2Se_5$ target is nearly stoichiometric, which is consistent with the conjectures discussed above. Intuitively, it is rather straightforward to explain why the $Bi_2Se_3$ target would lead to Bi-rich (or Se-deficient) film by recognizing that the re-evaporation of Se from the heated substrate (~350 °C) is much faster than Bi owing the much higher vapor pressure of Se [9,41]. The present results also suggest that to obtain stoichiometric $Bi_2Se_3$ films, a Se-excessive target is essential. We note that stoichiometric $Bi_2Se_3$ and $Bi_2Te_3$ films have been shown to exhibit reduced carrier concentration and increased carrier mobility, which led to the enhanced thermoelectric properties and provided suitable conditions for investigating the topological surface states [9,30,42].

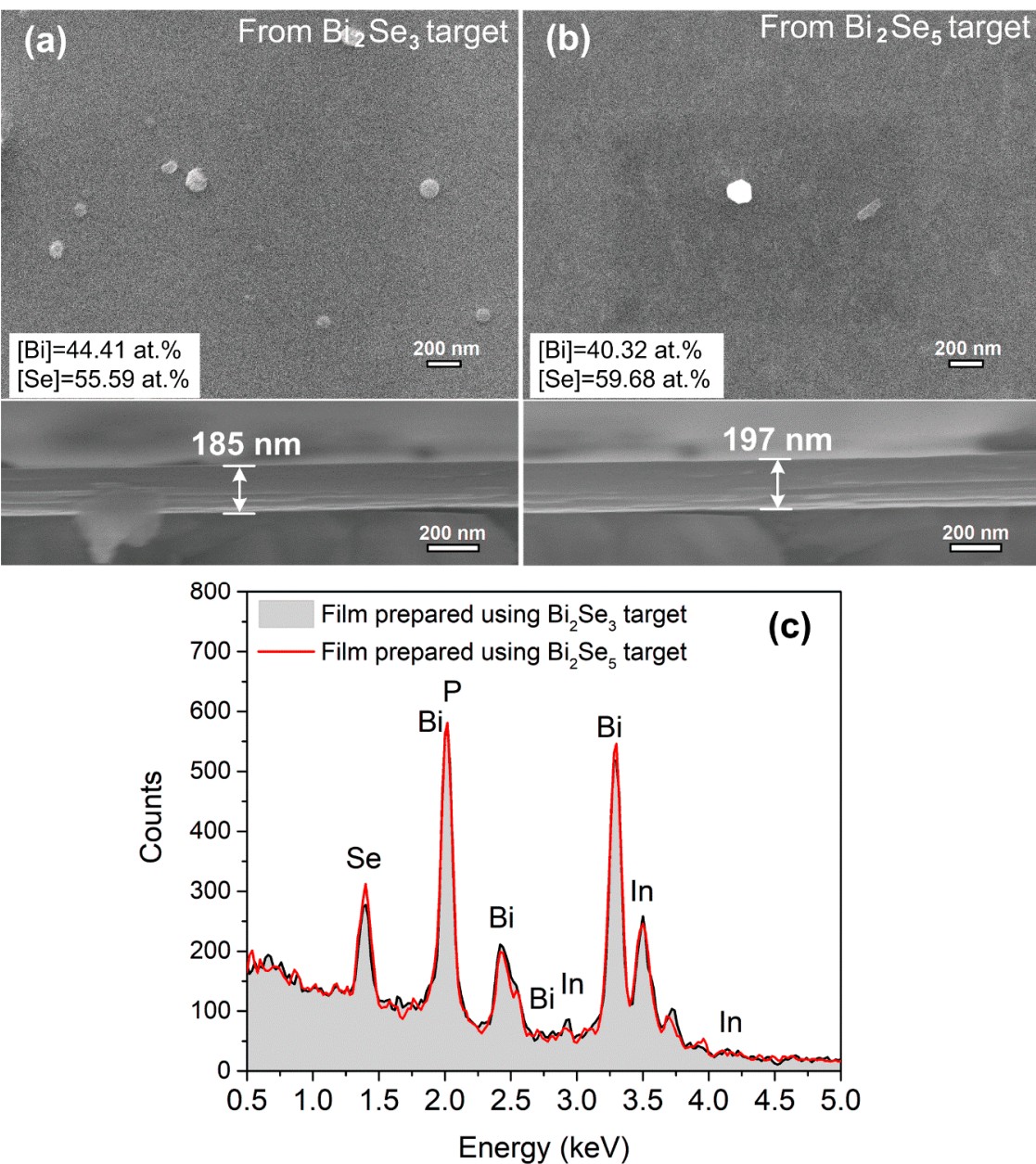

**Figure 3.** Top-view and cross-sectional SEM images of the Bi$_2$Se$_3$ thin films deposited from (**a**) Bi$_2$Se$_3$ target and (**b**) Bi$_2$Se$_5$ target. (**c**) EDS spectra of the corresponding Bi$_2$Se$_3$ thin films.

*3.2. Nanomechanical Properties*

The typical nanoindentation load–displacement curves of Bi$_2$Se$_3$ thin film deposited on InP(111) substrates are shown in Figure 4a. The hardness and Young's modulus of Bi$_2$Se$_3$ thin films were calculated from the load–displacement curves [39]; the Poisson's ratio of Bi$_2$Se$_3$ films is set to 0.25 in this study. Figure 4b,c present the penetration depth dependence of hardness and Young's modulus are obtained using the CSM method. In 2004, Li et al. [15] indicated that nanoindentation depth should never exceed 30% of the film's thickness. In this work, the CSM technique system is applied to record stiffness data along with load and displacement data dynamically, making it possible to calculate the hardness and Young's modulus at every data point and get their average values during the indentation experiment [15,39]. The mechanical properties obtained under nanoindentation exhibit a convergent manner and are steady with a rational tolerance around penetrating depths of 40~60nm, reflecting that the material properties obtained are intrinsic and the substrate effect on the present thin films for

hardness and modulus tests is negligible. The obtained values of hardness (*H*) and Young's modulus (*E*) are listed in Table 1 together with those reported in the literature for $Bi_2Se_3$ single crystals and thin films deposited on sapphire substrates.

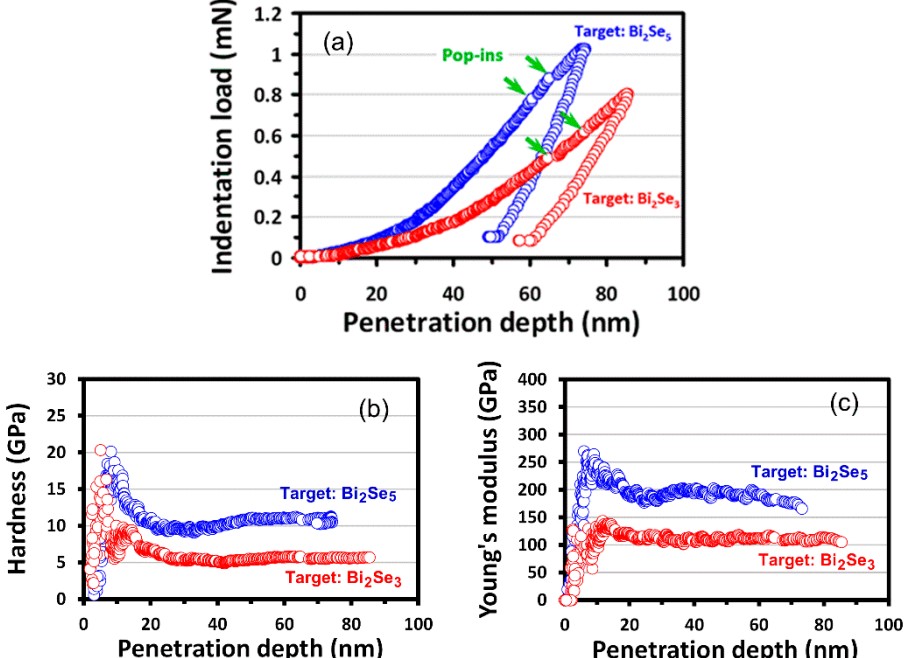

**Figure 4.** (**a**) The load–displacement curves of $Bi_2Se_3$ thin films deposited on InP(111) substrates using two different target compositions. A clear single "pop-in" behavior is displayed in both curves during loading. (**b**) A hardness—displacement curve and (**c**) a Young's modulus–displacement curve for a $Bi_2Se_3$ thin films deposited using $Bi_2Se_3$ and $Bi_2Se_5$ targets.

From Table 1, it is somewhat surprising to observe that the values of hardness and Young's modulus of the $Bi_2Se_3$ thin films are much larger than those of single crystals. The reason for this peculiar observation, especially the very low values for single crystals, is not clear at present. However, by comparing the results for films, the two prominent mechanical property parameters appear to have intimate correlations with the grain size (*D*) and surface roughness ($R_a$). For films grown on InP(111) substrate, as in the present case, the lattice mismatch between the $Bi_2Se_3$ thin films and substrate is about 0.2% [35], which, in turn, consistently resulted in films with better crystallinity, as indicated by the narrower full width at half maximum of the diffraction peaks, namely ~0.3° for films grown on InP(111) as compared to that of ~0.5° for the films grown on sapphire substrate [14]. Moreover, when comparing the results for the films grown with different targets, it further indicates that stoichiometry of the film can play an even more prominent role in determining the mechanical properties. Namely, the hardness and Young's modulus of the stoichiometric $Bi_2Se_3$ thin films are both about two times larger than that of Se-deficient films, which are again about two times larger than that grown on sapphire substrate. The enhancement of *H* and *E* values can be explained by considering the film crystallinity and surface roughness. It has been reported that the crystallinity of $Bi_2Se_3$ thin films deposited on InP(111) substrate was better than those deposited on $Al_2O_3$ and Si substrates [35]. In general, better film crystallinity often results in superior nanomechanical properties [43,44]. Therefore, compared with those reported in [14], the larger values of hardness and Young's modulus of the present $Bi_2Se_3$ thin films could be attributed to their better crystallinity. Furthermore, the film surface roughness can also be an important factor. Jian et al. [45] reported that the nanomechanical properties of ZnO thin films were significantly enhanced as the film surfaces became smoother. Even for AISI 316L stainless steel, the mechanical properties were found to decrease with increasing surface roughness [46]. Since the surface roughness

of the present films are all below 2.41 nm, it is reasonable to account, at least partially, for the enhanced *H* and *E* values.

Turning to the deformation behaviors during nanoindentation, it is evident that there are several pop-ins occurring along the loading segment for both load–displacement curves shown in Figure 4a. It is noted that similar phenomena were found in the previous studies [13,14], where the pop-ins were also observed in nanoindented $Bi_2Se_3$ single-crystal and thin films, despite the fact that the loads at which the pop-ins took place varied in each individual measurement. Moreover, it is noted that there is no sign of reverse discontinuity in the unloading portion of the load–displacement curves (the so-called "pop-out" event) being observed. The reverse discontinuity is commonly ascribed to the pressure-induced phase transformation that has been observed in Si or Ge single crystals [47,48]. The absence of these incidences indicates that pressure-induced phase transition did not occur for the $Bi_2Se_3$ films in the pressure range applied in this study. In fact, Yu et al. [49] have reported that the pressure-induced phase transition in $Bi_2Se_3$ occurred at pressures of 35.6 and 81.2 GPa as revealed, respectively, by Raman spectroscopy and synchrotron XRD experiments conducted in a diamond anvil cell. These values are much higher than the room-temperature hardness of the present hexagonal $Bi_2Se_3$ thin films. On the other hand, the pop-in behaviors during nanoindentation have been reported previously in other hexagonal structured materials, such as sapphire [50] and ZnO single crystals [51], as well as GaN thin films [52–54] by using the Berkovich indenter tip. It is generally conceived that the nanoindentation-induced deformation mechanism in these hexagonal-structured materials were primarily dominated by the nucleation and/or propagation of dislocations. Thus, it is plausible to believe that similar mechanisms must have been prevailing in the present $Bi_2Se_3$ thin films. Reasonably, it can be seen from Table 1 that the hardness of $Bi_2Se_3$ thin films increases when *D* value decreases, partially due to grain boundary hardening.

Within the context of the dislocation-mediated deformation scenarios, the first pop-in event may reflect the transition from perfectly elastic to plastic deformation. Namely, it is the onset of plasticity in $Bi_2Se_3$ thin films. Under this circumstance, the corresponding critical shear stress ($\tau_{max}$) under the Berkovich indenter at an indentation load, $P_c$, where the load–displacement discontinuity occurs, can be determined by using the following relation [55]:

$$\tau_{max} = 0.31\left(\frac{6P_cE^2}{\pi^3R^2}\right)^{1/3}$$ (1)

where *R* is the radius of the tip of nanoindenter. The obtained $\tau_{max}$ values are 1.8 and 3.4 GPa for $Bi_2Se_3$ thin films grown using $Bi_2Se_3$ and $Bi_2Se_5$ targets, respectively. The $\tau_{max}$ is responsible for the homogeneous dislocation nucleation within the deformation region underneath the indenter tip.

### 3.3. Wettability Behavior

The surface wettability of the $Bi_2Se_3$ thin films was examined by water contact angle measurements. If the contact angle ($\theta_{CA}$) is greater than 90°, it is said to be hydrophobic, otherwise it is hydrophilic. In Figure 5, the values of $\theta_{CA}$ for films are 80° and 110° for films grown using the $Bi_2Se_3$ target and the $Bi_2Se_5$ target, respectively.

As described above, the surface roughness measured by the AFM indicated that the $Bi_2Se_3$ thin film grown using the $Bi_2Se_5$ target have smaller surface roughness, suggesting that the wettability behavior of the surface was significantly affected by the surface morphology of the films [56]. Alternatively, the atomic arrangements and existence of surface defects might also play a role in the eventual surface energy. In general, the surface wettability is a measurement of surface energy and is most commonly quantified by $\theta_{CA}$ [57]. The surface energy for $Bi_2Se_3$ thin films was calculated by means of the Fowkes–Girifalco–Good (FGG) theory [58]. According to the analysis of the FGG method,

the considered critical interaction is the dispersive force or the van der Waals force across the interface existing between the water droplet and the solid surface. The FGG equation is given as:

$$\gamma_{ls} = \gamma_s + \gamma_l - 2\sqrt{(\gamma^d)_s + (\gamma^d)_l} \tag{2}$$

where $(\gamma^d)_s$ and $(\gamma^d)_l$ are the dispersive portions of surface tension for the solid and liquid surfaces, respectively. By combining Young's equation [56] with Equation (2) and taking the nonpolar liquid deionized water as the testing liquid and set $(\gamma^d)_l = \gamma_l$, the Girifalco–Good–Fowkes–Young equation becomes as: $(\gamma^d)_s = \gamma_l(cos\theta_{CA} + 1)/4$, where $(\gamma^d)_s$ is the surface energy of measured materials. Using $\gamma_l = 72.8$ mJ/m², the values of surface energy obtained were 21.4 mJ/m² and 11.9 mJ/m² for films grown with the $Bi_2Se_3$ target and $Bi_2Se_5$ target, respectively. The lower surface energy gives rise to higher hydrophobicity. It is noted that the $\theta_{CA}$ of 110° for the present stoichiometric $Bi_2Se_3$ thin films deposited on InP(111) substrates using PLD is even larger than that ($\theta_{CA} \sim 98.4°$) of $Bi_2Se_3$ thin films deposited on $SrTiO_3$(111) substrate by MBE [59]. In any case, the present study suggests that both the hydrophobic/hydrophilic transition behavior and nanomechanical properties of the $Bi_2Se_3$ thin films can be manipulated by controlling the target compositions.

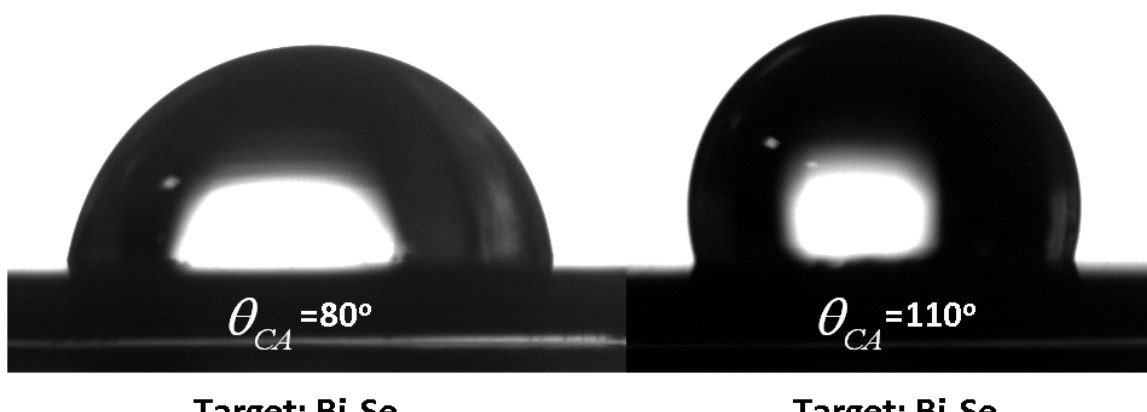

**Figure 5.** Contact angle test: the images of water droplets on the $Bi_2Se_3$ thin film surfaces.

## 4. Conclusions

The present study evidently illustrated that stoichiometry, which can be manipulated by tuning the target composition, can give rise to significant effects on the microstructural, morphological, compositional, nanomechanical and surface wetting properties of the $Bi_2Se_3$/InP (111) thin films. The $Bi_2Se_3$ thin films were grown using PLD from a stoichiometric $Bi_2Se_3$ target and a Se-rich $Bi_2Se_5$ target at a substrate temperature of 350 °C in a vacuum with a base pressure of $\sim 4 \times 10^{-6}$ Torr. The films were highly (00l)-oriented with smooth surfaces consisting mainly of triangular step-and-terrace structures, which is the common feature of epitaxial $Bi_2Se_3$ thin films. Compared to the films grown from the $Bi_2Se_3$ target, using the $Bi_2Se_5$ target is more favorable for obtaining stoichiometric films with larger hardness and Young's modulus. In addition, the contact angle (surface energy) of the $Bi_2Se_3$ films deposited from the $Bi_2Se_3$ and $Bi_2Se_5$ targets were 80° (21.4 mJ/m²) and 110° (11.9 mJ/m²), respectively. These results suggest that, in addition to the usual factors such as surface roughness and grain morphology, stoichiometry as well as defect chemistry originated from Se-deficiency may also play important roles in determining the eventual nanomechanical and wettability properties of $Bi_2Se_3$ thin films.

**Author Contributions:** Data curation, Y.-M.H., C.-T.P., B.-S.C., P.H.L., L.T.C.T., N.N.U. and V.N.; Formal analysis, Y.-M.H., C.-T.P., B.-S.C., L.T.C.T., N.N.U. and V.N.; Funding acquisition, J.-Y.J.; Resources, P.H.L., C.-W.L., J.-Y.J.,

J.L. and S.-R.J.; Writing—original draft, P.H.L. and S.-R.J.; Writing—review & editing, J.-Y.J. All authors have read and agreed to the published version of the manuscript.

**Funding:** This research was funded by the Ministry of Science and Technology, Taiwan under Contract Nos. MOST 109-2221-E-214-016.

**Acknowledgments:** The authors would like to thank T.-C. Lin for her technical support in the nanoindentation experiments.

**Conflicts of Interest:** The authors declare no conflict of interest.

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
