# Peer review of "Effects of Stoichiometry on Structural, Morphological and Nanomechanical Properties of Bi2Se3 Thin Films Deposited on InP(111) Substrates by Pulsed Laser Deposition"

_coatings, doi:10.3390/coatings10100958_

Round 1

Reviewer 1 Report

In this article the Authors report a study on the effect of stoichiometry of bismuth selenide coating on InP(111) substrate, obtained by pulse laser deposition, focusing on the structural, morphological, mechanical and surface wetting properties. The thin films were characterized by means of XRD, SEM-EDS and AFM, moreover the hardness and Young’s modulus were measured as well as the wettability.

In general, this article is well written, the research design is appropriate, and the English is good, the results are well discussed, and the conclusions are consistent with the reported data. Although the level of novelty is not high, the study clearly shows the effect of the stoichiometry.

Some suggestions follow:

  1. The title refers to the effect of the stoichiometry of Bi2Se3, but the stoichiometry of Bi2Se3 is demined, so it may be more appropriate Bi2SeX.
  2. Line 108: there is an error.
  3. Line 120: “impurity”. The impurities may not be detectable by the XRD and in any case it should be specified which type of impurity you are talking about.

Author Response

Dear Reviewer,

Please kindly find the attached file.

Jian

Reviewer 2 Report

The work concerns the microstructure and selected properties of thin Bi2Se3 layers deposited by the PLD method on InP (111) substrates, using two targets of Bi2Sex ( x=3, 5), different in stoichiometry with respect to Se content. The work was created with the participation of 11 authors. Presented examinations and results are basic, include microstructure analysis of deposits using XRD, (unknown geometry, probably Bragg-Brentano), SEM (with EDS) and AFM, as well as nanoindentation test and wettability tests.

the reviewer believes that the manuscript is not fully ready for publication and requires supplementation.

Remarks and comments.

1.Targets- what was the phase composition of targets, their hardness and Young modulus,  how their surfaces were prepared for PLD process? What was the geometry of pulsed laser deposition?

2. Fig.1. Please add XRD spectra registered for targets, and specify standard patterns numbers (add maxima positions for standard XRD pattern) for identified phases.

The shift of the maxima position does not have to be a result
of a change in the stoichiometry of the thin film but, for
example, of difference in internal stress biult up during deposition.

3. Figs 2a and b- poor quality, illegible details, pink areas on Fig 2a? please unify scale bar number size

I strongly recommend to change 2D images into 3D (XYZ) ones with scale bares.

Fig 2. c and d, specify how the samples for SEM observations were prepared. EDS insterts are too small, maxima indexes are illegible. Please specify ( or mark)  areas from which EDS spectra were taken and explain the origin of registered elements (except for Bi and Se)

 The accuracy of the chemical composition analysis by EDS is 0.1%

4. Fig.3- For nanoindentation tests in CSM mode, please give plots hardness vs load/penetration depth and Young modudlus vs load/ penetration depth.

What values of the Poisson ratio were taken into calculations? Maximum penetration depth to the thin film thickness ratio has to be 10 % or less, so that the values can be interpreted as not affected by substrate properties. Recommended above plots will allow us to assess this.

5. Table 1 what is D? and why this value was chosen? What was the hardness and Young modulus for InP substrate determined in the same test conditions?

6. The conclusions are not transparent and require redrafting.

They should not contain so many parameters of conducted experiments.

Author Response

(The authors gave the same response as above.)

Reviewer 3 Report

Dear Authors, Please check typos and revise English : 

-line 108 : 10 m apart

-line 128: the films are.

Section 3 should be Results and discussion

Author Response

(The authors gave the same response as above.)

Round 2

Reviewer 2 Report

Remarks.

1. Fig.1 - please add powder diffraction file standards (spectra or mark peak possitions) for Bi2Se3 and InP and give numbers of standard cards for identified phases (in the figure description).

There are some undescribed maxima on both of plots ( at 2 theta: ~28, 29.5 dublet, at 35°, 45° and 57° at “b” plot; 34°, 39° and  56° at “a”  plot ). PLD deposits are usually nonequilibrium, multiphase, please explain why only Bi2Se3 phase is identified. Strong and complex maximum positioned at 2 theta of 27-28° could be identified as BiSe (014) or Bi2Se3 (015) while maximum at ~54° could be also registered for BiSe (024).

Please indicate the type of lattice and lattice parameters for identified phase and compare them to standard ones ( in the Table).

2. Could pop-in events be the effect of decohesion in deposited thin films or loss of adhesion to InP substrate (by its plastic deformation and subsequent cracking at the  substrate- thin film boundary, or by cracking in InP substrate as the effect of absorbing excess strain energy during multiple loading during CSM nanoindentation test?

 3.  In the nanoindentation tests, the examined system is soft film on hard substrate, as for Bi2Se3  referred hardness (determined in nanoindentation tests) is  ~200 MPa, E ~ 4GPa [https://doi.org/10.1016/j.physb.2020.412275]) and for InP substrate ( H ~7GPa, E ~78 GPa [S.-R. Jian, J.S.-C. Jang / Journal of Alloys and Compounds 482 (2009) 498–501]), so during nanoindentation with indentation depth (30-80 nm) to film thickness ( 199 nm) ratio is 15-40 %. Taking into account the referred values, it is obvious that mechanical properties of the InP substrate influenced hardness and Young modulus values calculated by Authors.  What is the other explanation of very high values of calculated hardness and Young modulus  of deposits? [see for exampleA. Iost et al. / Thin Solid Films 524 (2012) 229–237].

4. Table 1, Please add the hardness and E values for substrates (as referential data).

Author Response

Dear Reviewer,

Thank you,

Jian

Round 3

Reviewer 2 Report

Authors explained all ambiguities, took into account the comments and
suggestions of the reviewer. The corrections have been made.
The manuscript in its present form is eligible for publication in Coatings.